# Sleep Disorders and Mental Stress of Healthcare Workers during the Two First Waves of COVID-19 Pandemic: Separate Analysis for Primary Care

**DOI:** 10.3390/healthcare10081395

**Published:** 2022-07-26

**Authors:** Athanasia Pataka, Seraphim Kotoulas, Asterios Tzinas, Nectaria Kasnaki, Evdokia Sourla, Evangelos Chatzopoulos, Ioanna Grigoriou, Paraskevi Argyropoulou

**Affiliations:** 1Medical School, Aristotle University of Thessaloniki, 54124 Thessaloniki, Greece; akiskotoulas@hotmail.com (S.K.); stergiostzinas@hotmail.com (A.T.); evisou@yahoo.gr (E.S.); vaggchatz@hotmail.com (E.C.); pargyrop@auth.gr (P.A.); 2Respiratory Failure Unit, G. Papanikolaou Hospital, 57010 Thessaloniki, Greece; kasnakinek@gmail.com (N.K.); ioagrig@hotmail.gr (I.G.)

**Keywords:** sleep disorders, COVID 19, waves, primary care, sleep condition indicator, anger, anxiety, depression, PHQ4, DAR-5, loneliness

## Abstract

Background: During the recent pandemic, Healthcare Professionals (HCPs) presented a significant prevalence of psychological health problems and sleep disturbances. The aim of this study was to assess the impact of COVID-19 on HCPs’ sleep and mental stress with a separate analysis for primary care HCPs. Methods: A cross-sectional observational study with an online anonymized, self-reported questionnaire was conducted in May 2020 (1st wave) and repeated in December 2020 (2nd wave). Patient health questionnaire-4 (PHQ-4), dimensions of anger reactions-5 (DAR-5) scale, 3-item UCLA loneliness scale (LS) and sleep condition indicator (SCI) were used. Results: Overall, 574 participants were included from the 1st wave, 514 from the 2nd and 469 were followed during both. Anxiety and depression were significantly higher during the 2nd wave vs. the 1st (32.8% vs. 12.7%, *p* < 0.001 and 37.7% vs. 15.8%, *p* < 0.001). During the 2nd wave, HCPs scored significantly higher in DAR-5 (9.23 ± 3.82 vs. 7.3 ± 3.3, *p* < 0.001) and LS (5.88 ± 1.90 vs. 4.9 ± 1.9, *p* < 0.001) with worse sleep quality SCI (23.7 ± 6.6 vs. 25.4 ± 3.2, *p* < 0.001). This was more evident in primary care HCPs. Significant correlations were found between SCI and PHQ4, DAR5 and LS. Conclusion: There is a need to support HCPs’ mental health and sleep, especially in those working in primary care.

## 1. Introduction

In December 2019, a novel coronavirus was identified (SARS-CoV-2) leading to a global pandemic and subsequently increasing the amount of pressure on Healthcare Professionals (HCPs). HCPs were exposed to situations outside their ordinary work and experienced high levels of stress and irregular schedules with frequent and long work shifts [1,2,3]. During the previous epidemics, such as H1N1, SARS and MERS, HCPs reported significant levels of physiological distress and sleep disorders and it has been shown that disturbed sleep is closely associated with anxiety, depression and post-traumatic stress [4,5,6]. During the recent pandemic, HCPs presented a significant prevalence of psychological health problems, burnout and sleep disturbances, especially insomnia [7,8,9,10,11,12,13,14,15]. Sleep disorders are a common problem also reported in the general population during the pandemic with prevalence similar or lower than that of HCPs. However, significant variations were observed in the literature with estimates for sleep problems among different populations ranging from 8% to 91% [16]. Patients infected from SARS-COV-2 appeared to be the most affected group with a prevalence reaching 75% [16]. Furthermore, in a recent systematic review with a meta-analysis, the rates of sleep problems were found to be comparable in both HCPs and the general population (i.e., 36% in HCPs vs. 32% in the general population) [16]. Despite the variation between different systematic reviews and meta-analysis, the pooled prevalence of sleep disorders in HCPs ranged from 36% to 45% [9,16,17].

HCPs are considered as pivotal to COVID 19 crisis management, playing a major role in screening and treating patients, especially HCPs working in primary care. Furthermore, at the beginning of the COVID 19 pandemic, the failure of providing essential care protective equipment, the rapid transmission of the disease and the lack of a definite treatment increased the feeling of insecurity, powerlessness and frustration which further increased psychological and mental health problems. Increased pressure during work results in an increased risk of adverse mental health outcomes and also sleep disorders [11,12,13,14,15,16,17,18,19,20]. Lifestyle factors such as smoking and alcohol use, the use of substances and the pre-existence of mental health disorders may influence the development of subsequent mental health conditions and sleep disorders in HCPs [21,22,23].

During the pandemic, the role of primary care, and primary care HCPs became more important as primary care centers were the first point of contact for patients suspected of COVID-19; additionally, the majority of hospitals were transformed into COVID 19 pandemic facilities. Primary care HCPs served in the diagnosis and treatment of suspicious or diagnosed COVID 19 cases in the hospitals, especially in the emergency departments. This led to an increase in the workload but also in the risk of primary care HCPs’ exposure to COVID 19. The effect of the pandemic on the mental health and sleep of frontline HCPs has been proven to be significantly worse compared with non-frontline HCPs [24].

As the pandemic evolves, its effects on the mental health of HCPs may change. Notably, the majority of studies so far have been web-based and several of them were performed during 2020, i.e., before the initiation of vaccination [9,10,11,25,26]. Most of the existing research examines the impact of COVID-19 on sleep using cross-sectional surveys with questions addressing possible changes in sleep and an evaluation of changes from one time point to another (between different waves of the pandemic or before and after different lock downs) [27,28].

We hypothesized that, after experiencing the first COVID-19 wave, HCPs’ level of sleep quality together with psychological distress, such as anxiety and depression, anger and loneliness, further deteriorated, especially in primary care HCPs more exposed to COVID 19. We aimed to compare the same groups of HCPs during the two first consecutive waves and to evaluate if demographic and social factors, exposure to COVID 19 and psychological distress affected sleep quality with a separate analysis for primary care HCPs.

## 2. Methods

This was a cross-sectional time-series study (https://dam.ukdataservice.ac.uk/media/455362/changeovertime.pdf (last accessed 8 July 2022)). The protocol was approved by the institutional review board of G Papanikolaou General Hospital, Exohi, Thessaloniki, Greece (N 876, Approval Date 20 May 2020), and all the participants provided their written electronic consent that was included in the online questionnaire before the initiation of enrolment. An information sheet with detailed explanations of the research purpose was included in the survey. For several questions, the participants were not able to continue to the next question of the survey unless they had provided a response. The participants were informed about the anonymization of the data and no incentives were offered. A pilot study was conducted from a sample of 30 HCPs working in G Papanikolaou Hospital before the initiation of data collection in both waves and these results were excluded from the survey.

The questionnaire of the study was sent to all employees of the hospitals and healthcare centers (physicians, nurses, physiotherapists, etc.) of both the public and private sector in the province of Thessaloniki, Greece, via emails delivered from the local medical and nurses associations during the first two epidemic waves of COVID-19; the first during May 2020 and the second during December 2020. The questionnaire included questions about the baseline characteristics of the participants (age, gender, marital status, children, education, occupation, job location, specialty, work in an intensive care unit (ICU) and with COVID patients), as well as alcohol consumption, smoking habits assessed by the Heaviness of Smoking Index (HSI) [29], anxiety and depression using the patient health questionnaire-4 (PHQ-4) [30], anger using the dimensions of anger reactions-5 (DAR-5) scale [31], loneliness using the 3-item UCLA loneliness scale [32], and sleep related disorders using the sleep condition indicator (SCI) [33]. During the follow up of the HCPs (second wave), the questionnaire included a question asking the participants to declare whether they had completed the questionnaire previously. From this question and by crosschecking with the demographics of the participants, we managed to maintain the same groups of HCPs that were followed up during both waves and measured the variables in the same respondents over time.

The HSI is a questionnaire that evaluates nicotine dependence by using only the following two items: the time to first cigarette in the morning and the number of cigarettes per day. Scores ranging from 0 to 2 indicate low addiction, from 3 to 4 moderate addiction and from 5 to 6 high addiction [29].

The PHQ4 [30] is a brief screening scale for the evaluation of anxiety and depression consisting of 4 questions rated from 0 to 3. The total score of PHQ4 is determined by the addition of each of the 4 questions together with scores from 0–2 rated as normal, 3–5 as mild, 6–8 as moderate and 9–12 as severe. If the total score of the first two questions is ≥3, anxiety is suggested, whereas if the total score of the last two questions is ≥3, depression is suggested [30].

The DAR-5 [31] is a 5-item self-report five-point Likert scale, ranging from 1–5, which is used for the assessment of anger intensity, duration, frequency and impact on social functioning over a 4-week period. The total score ranges from 5 to 25 with higher scores indicating worse symptomatology. A cut-off score of above 12 is indicative of functional impairment and psychological distress related to anger [31].

The UCLA loneliness scale comprises of three questions that evaluate three dimensions of loneliness (relational and social connectedness and self-assessed isolation). The scores for each question are added together with a range of scores from 3 to 9; with individuals with scores of 6–9 characterized as “lonely” [32].

The SCI [33] is an eight-item scale, based upon DSM-5 insomnia criteria, comprising two items evaluating sleep continuity, two items evaluating sleep satisfaction or dissatisfaction, two items evaluating the severity of the sleep disorder (nights per week; duration of problem) and two evaluating the daytime consequences of poor sleep (effects on mood, energy or relationships, concentration, daytime performance). Each item is scored on a five-point scale (0–4), with lower scores (ranging from 0 to 2) reflecting DSM-5 threshold criteria for insomnia disorder. Total SCI scores ranged from 0 to 32, with higher values indicating better sleep. An SCI score of ≤16 was found to be indicative of possible insomnia disorder [33,34].

## 3. Statistical Analysis

Statistical analysis was performed using SPSS (version-20 IBM-SPSS-statistical-software, Armonk, NY, USA). Continuous variables were presented as mean ± SD and categorical variables as number (%). *p* <0.05 was accepted as statistically significant. The normality of continuous data was assessed using the Kolmogorov–Smirnov test. For the detection of statistically significant differences between the whole population of responders of the two waves, independent-samples-T-test for parametric and the Mann–Whitney-U-test for non-parametric variables were used. Paired *t*-test or McNemar’s test were used to compare the same HCPs that were followed up during both the first and second waves for parametric and non-parametric variables, respectively. A chi-square was used for the detection of significant differences between the two waves in categorical variables. Regression analyses were performed to evaluate the association of predictive factors with changes in sleep quality across the two time points, whereas Pearson’s Correlation was used to explore the associations of different of factors (SCI, PHQ4, DAR5, LS) in the two different waves. Variables including age, gender, employment status (working in primary care or not, working in the private sector), depression, anxiety, anger and loneliness were included. Associations between different risk factors and poor sleep quality, assessed by SCI cut off 16, were determined with binary logistic regression analysis. The identified covariates were presented in terms of correlation coefficient (r or R2). Additionally, a separate analysis was performed only for primary care HCPs. G*Power software was used to calculate the sample size showing that a sample size of 248 would be required for an expected effect size of d = 0.55, an alpha of 5%, and beta of 20% for two groups with an allocation ratio of 1.8 when performing Mann–Whitney U tests.

## 4. Results

Five hundred and seventy-four participants answered the questionnaire during the first wave, 514 during the second wave and 469 were followed during both the waves. The socio-demographic characteristics of the participants of the two groups were comparable, apart from the absence of physiotherapists among the participants from the second wave (Table 1). It was also evident that during the second wave, the participants had been treating COVID-19 patients more often than in the first (*p* < 0.001). According to the answers of the PHQ-4 questionnaire, HCPs presented significantly worse mental status during the second wave with more anxiety and depression symptoms (Table 1). Additionally, the participants felt angrier and lonelier compared to the first wave. They scored significantly higher in the DAR-5 and the UCLA loneliness scale (Table 1).

When the mental health variables and sleep quality of the 469 respondents that were followed up during both first and second waves were compared, worse outcomes were found in all parameters. Sleep quality, anxiety, depression, anger and loneliness were significantly worse during the second wave of the pandemic (Table 2)

During the second wave, the participants presented significantly worse sleep quantity (*p* < 0.001) and quality (*p* < 0.001) with more awakenings compared with the first wave with 46.8% reporting that they experienced problems with their sleep 2 ≥ nights per week (Table 3). Additionally, 16.3% of HCPs, compared with 5.7% in the first wave, thought that poor sleep affected their mood, energy or relationships during the past month, whereas 11.7% vs. 5.5% thought that poor sleep affected their productivity, concentration and ability to stay awake (Table 3). Almost 11% of HCPs during the second wave believed that poor sleep caused trouble to them in general, compared with 4.3% in the first (Table 3). During the second wave, a significantly lower total score in the SCI (23.7 ± 6.6 vs. 25.4 ± 3.2, *p* < 0.001) was found, a fact that meant more frequent insomnia symptoms with a SCI cut off 16 (18.6% vs. 6.1%, *p* < 0.001) (Table 2).

In a separate analysis of primary care HCPs, there were statistically significant differences between them and all the other HCPs, as they were younger, mostly female, and experienced a worse quality of sleep, more anxiety, anger and loneliness (Table 4). These differences were more evident during the first wave of the pandemic. During the second wave, no significant differences were observed between the groups of HCPs, with similar scores in SCI, PHQ4, DAR5 and LS. However, when the sleep quality, depression, anxiety, anger and loneliness of each group of HCPs, i.e., primary care HCPs or all the others, were compared between the two waves, significant aggravation was found in almost all variables and in both groups (Table 4).

In the regression analysis, the predictors that mostly affected sleep quality during the first wave were being a primary care HCP, being married and psychological distress assessed by PHQ4. However, during the second wave, the most important factors were again psychological distress, anger and loneliness, but not marital status and being a primary care HCP. Parenthood did not associate with sleep quality as assessed with SCI. Treating COVID patients did not have a significant effect on the quality of sleep of HCPs. In the separate analysis of primary HCPs, the most significant factor that affected sleep quality during the first wave was PHQ4, and during the second wave additionally DAR5 and loneliness (Table 5). Treating COVID 19 patients, parenthood and marital status were not associated with the sleep quality of primary care HCPs in the regression analysis during both waves.

In a separate gender analysis, the multiple regression analysis (age, primary care or not, PHQ4, DAR5, LS, HIS) revealed that apart from PHQ4, treating COVID patients (β = −0.175, *t* = −2.7, *p* = 0.008) and DAR5 (β = −0.16, *t* = −2.04, *p* = 0.04) were significantly associated with sleep quality (SCI) in men but only during the first wave. During the second wave in men, and in both waves in women, the associations did not differ from the analysis for both genders together (Table 5). The only factor associated with SCI cut off 16 was PHQ4 in men (OR: 1.94, 95% Confidential Interval 1.13–3.31, *p* = 0.015), but not women (OR: 1.6, 95% Confidential Interval 0.64–4.01, *p* = 0.21) and only during the first wave.

In the further analysis of psychological distress assessed by the PHQ4, marital status and parenthood did not associate with anxiety and depression. PHQ4 significantly associated with SCI, DAR5, Loneliness (*p* < 0.001) and female gender (*p* < 0.04) in both waves. Significant negative correlations were found between SCI and PHQ4, DAR5 and LS for all the participants in both waves and also in the separate analysis for primary care HCPs (Table 6).

## 5. Discussion

To the best of our knowledge, this is the first study that attempted to demonstrate the changes in sleep quality, mental distress, anger and loneliness in Greek HCPs in relation to the evolving COVID-19 pandemic. To date, no longitudinal studies evaluating these factors on HCPs and especially primary care HCPs have been conducted in northern Greece. Our results indicate that the sleep quality of all HCPs significantly worsened during the pandemic, together with the levels of anxiety, depression, anger and loneliness. Primary care HCPs were more significantly affected in both waves.

There is a bidirectional relationship between psychological distress and sleep quality. Anxiety and stress affect sleep quality by being the main precipitating factors for the development of sleep dysfunction and, on the other hand, sleep quality is an important factor for the regulation of stable emotional control [35,36,37,38]. In a meta-analysis evaluating the psychological and mental impact of COVID 19 on medical staff and the general population, the prevalence of anxiety was found to be rather similar between these groups (26 % and 32%, respectively) [37]. The COVID-19 pandemic is the most recent global public health event after SARS and Middle East Respiratory Syndrome (MERS). There is evidence that HCPs suffered from emotional distress and psychiatric morbidity during all these previous outbreaks with prevalence reaching almost 50% [39,40].

During the early stage of the COVID-19 pandemic, anxiety symptoms were reported in 28.8% and depressive symptoms in 16.5% of the general population in China [41]. On the other hand, among HCPs the incidence of anxiety and depression was reported to be around 44% and 50%, respectively [42,43]. In our study, HCPs were found to have a significantly worse mental status during the second wave compared to the first, with anxiety symptoms reported in 32.8% and depression in 37.7% of the participants. This was more evident in primary care HCPs compared with the other groups of HCPs with 23.8% reporting symptoms of anxiety and 27.5% of depression. Anxiety symptoms were also found to be more frequent than depression in other studies [43]. The application of different tools to assess mental health, and using different cutoff points, may have contributed to the heterogeneity of results between different studies. Mild symptoms of fear and stress are considered a normal reaction to a newly recognized condition such as the COVID 19 pandemic. It seems that most HCPs experienced mild symptoms of both depression and anxiety, as in our study, while more severe symptoms were reported less frequently [9,43].

Considerable evidence, especially in the form of cross-sectional studies and systematic reviews, has demonstrated the significant impact of COVID 19 on psychological and mental health outcomes and sleep disorders [35,36,37,38,39,40,41,42,43]. However, until now, limited published studies were available evaluating the changes in the mental health of HCPs during the course of the pandemic and more specifically comparing the differences over the different waves. Similar to our findings, other studies [44] also reported higher odds of psychological distress, such as depression and anxiety, among those HCPs who were involved with the treatment of patients suffering from COVID19. Additionally, a longitudinal study that was carried out in Japan and evaluated four waves of the pandemic showed that even when the cases of COVID 19 were low, HCPs continuously experienced high psychological distress [45]. Our results indicate that the levels of anxiety, depression, anger and loneliness together with the worsening of sleep quality increased during the course of the pandemic. Several factors may influence mental health outcomes. Specifically, in our study, negative correlations were observed between psychological distress, anger, loneliness and sleep problems, especially in primary care HCPs, which makes more evident the importance of implementing strategies to improve the work environment and support HCPs.

The existing literature reported that the prevalence of psychological symptoms in HCPs during the COVID-19 pandemic was higher than in the previous epidemics [44,45,46,47,48,49,50,51]. We found that during the second wave, the participants felt angrier and lonelier compared to the first, with 22% presenting clinically significant anger and 54.3% indicating significant loneliness. This may have a negative impact on the provision of health services [52]. A survey evaluating the impact of COVID-19 on mental health and its associated factors among HCPs across 31 countries showed that more psychological consequences were found in the HCPs with less social support possibly because they did not have the opportunity to express their feelings [50]. This is in accordance with our study as loneliness was found to be an important factor affecting sleep quality especially during the second wave in the total population and separately in primary HCPs.

Stress is a well-known cause of sleep disturbances in HCPs [44,45,46,47,48,49,50,51] and possibly due to that, the prevalence of sleep disturbances in COVID-19 medical staff was higher than in other community groups [53]. In Wuhan, HCPs experienced high levels of depression anxiety, anger, fear and stress, due to the possibility of infection, the direct exposure to disease and excessive work pressure [51]. In addition, it was found that 39.2% of HCPs in China suffered from sleep disturbances during the pandemic [43]. In our study, a strong correlation was found between sleep quality (SCI) and mental health (PHQ4), even in the separate analysis of depression and anxiety, further verifying this important bidirectional relationship. This was even more evident in the separate analysis of primary care HCPs.

It was found that consultants and physicians with greater responsibility for treating COVID-19 patients developed sleep disturbance more frequently [19,50,54,55]. A systematic review and meta-analysis revealed that almost 35% of HCPs suffered from sleep disturbances, with those working at the front line against COVID-19 being more vulnerable. Increased stress in the workplace due to exposure to COVID 19 increased sleep problems in HCPs and especially nurses and physicians [56]. In our analysis, primary care HCPs presented worse sleep quality that was aggravated during the evolution of the pandemic in the second wave with 20.6% reporting symptoms of insomnia. During that time, almost 50% of HCPs that responded were providing treatment to COVID 19 patients.

On the other hand, a study during the early outbreak of COVID-19 in Hubei Province [57] demonstrated that the sleep problems (PSQI scores) of both frontline and non-frontline workers did not differ significantly; however frontline HCPs were more susceptible to more severe sleep disturbances (PSQI > 10). In that study, binary logistic regression analysis revealed associations between medical occupation, parenthood, anxiety and depression with poor sleep quality. In our study, the most significant associations were between being primary care HCP and psychological distress (PHQ4) during the first wave and between psychological distress, anger and loneliness during the second wave with poor sleep quality.

Other factors that may contribute to sleep problems in HCPs are being female [58,59,60], aged 41–45 years [51,60], caring for children [57,58] and being single [60]. Importantly, in our study female gender did not show an independent effect on SCI as well as parenthood and marital status. In previous studies, female sex was associated with poor sleep quality, increased anxiety and stress compared with men [61,62], and this also applied for HCPs during COVID-19 pandemic [52,56,63]. Additionally, it was found that sleep disturbances among HCPs positively related to increased age and this could be attributed to increased fatigue from work during the years, physical problems and increased need for rest and sleep with older age [56,63]. In our study, age did not affect the results probably because most of the HCPs were of the same age range (40–50 years old).

The majority of studies so far quantified sleep habits with the use of self-reported questionnaires such as the Pittsburgh Sleep Quality Index (PSQI), Athens Insomnia Scale (AIS), Insomnia Severity Index (ISI), and some also used questionnaires designed by the researchers [64]. The selection of the questionnaire may have a significant impact on the results and may possibly explain the heterogeneity of prevalence rates of sleep disorders across different studies [9,16]. For example, with the administration of PSQI, higher rates of sleep disturbances were demonstrated compared with AIS and ISI; this could be attributed to the fact that the PSQI evaluates sleep quality in general assessing a broad range of sleep disorders such as snoring, sleep medications and nightmares, whereas AIS and ISI are more specific to insomnia symptoms [16]. Based on scores such as the PSQI, the prevalence of poor sleep quality in HCPs ranged between 18.4 to 84.7% [16,43,55,61,64,65,66,67,68]. A study showed that HCPs’ sleep quality worsened after one-month during the early COVID-19 pandemic, with the percentage of HCPs presenting PSQI > 5 increasing from 62% to 69.3% [68]. In a rather recent study evaluating Italian pediatric HCPs, the median PSQI value was 8.0 (5.0–10.0) with 68.6% of the participants having a score higher than 5 indicating sleep disturbance, 20.0% of 11–16 indicating poor sleep quality and 2.3% more than 16 indicating very poor sleep quality [69]. In our study, we used the SCI for the assessment of sleep disorders with a cut off of 16 being indicative of insomnia disorder. During the progression of the pandemic, 18.6% of HCPs had a SCI score ≤16 which is indicative of a possible insomnia disorder. The deterioration of insomnia was evident in both primary care and non-primary care HCPs during the second wave pandemic with 20.6% of primary care HCPs and 14.3% of all the other HCPs having SCI score ≤16.

Insomnia was the most frequent sleep disorder found in other studies with a prevalence ranging from 23.6% to 68% when assessed by ISI scores, with ISI≥15 indicating moderate-to-severe insomnia in 7–15% [70]. Other studies using AIS with a cut-off of 6 found that 68% of physicians and 53% of nurses suffered from insomnia [69,70,71,72]. Apart for insomnia, other sleep disorders such as nightmares, sleep terrors and sleepwalking were more frequently reported in HCPs [66]. Furthermore, a study that evaluated HCPs insomnia using wearable sleep oximeters found a high rate of co-morbid moderate to severe sleep apnea in HCPs with insomnia being attributable to stress reaching up to 38.5% [73].

The main strength of our study lies on the cross-sectional time-series design to study differences in sleep quality and mental health symptoms among the same groups of HCPs during two consecutive waves of the pandemic. However, our study has some limitations. It was based on an online questionnaire that cannot guarantee the accuracy of all the information provided from the participants. Additionally, it was very difficult to overcome the selection bias as the study was based on an online survey and the sampling from each facility might not be representative. Unfortunately, the majority of the participants of our study were physicians, and this could be the reason that we did not find any significant differences between different types of HCPs, especially in nurses so we cannot make safe conclusions and comparisons between different groups of HCPs. Unfortunately, the existence of previous history of anxiety and/or depression before the pandemic was not evaluated and this could have affected our results. However, the significant deterioration of these symptoms over the pandemic is an important finding even in the absence of data of a possible prior disorder.

The relationship between COVID-19- related sleep disturbance and stress is bidirectional. Poor sleep may lead to daytime sleepiness, fatigue and impaired daytime performance resulting in work errors further worsening the psychological condition of HCPs. During a crisis such as COVID-19, the good quality of sleep of HCPs becomes of essence as poor sleep or sleep deprivation may reduce work efficiency by impairing cognitive functioning and decision-making processes, increasing the risk of medical errors and poor patient outcomes. In addition to poor patient outcomes, reduced sleep quality has been related to decreased personal satisfaction, adverse mental health and increased burnout and may also be associated with increased morbidity, risk for obesity, diabetes and cardiovascular complications such as heart attack and stroke [23]. Stress and depression have been linked with increased likelihood of impaired professional behavior [39,53]. The psychological impact of the pandemic is more evident in frontline HCPs, but it is also felt by HCPs of other specialties [9,16,69]. Sadly, the emotional impact of the pandemic has led to suicides among HCPs [66]; something that is very alerting as compared with the general population physicians are at an increased risk of committing suicide [74].

As sleep disorders, especially insomnia, have been related in the development of mood changes, and vice versa [36,59,75], a position paper referring to the protection of HCPs during the COVID-19 pandemic emphasized that apart from personal protective equipment and food, psychological and family support are also important [76]. The European Academy for Cognitive-Behavioral Treatment of Insomnia (the European CBT-I Academy) provided some recommendations addressing the sleep problems of HCPs who experienced an increased work burden during the COVID-19 pandemic. Practical recommendations such as expressing concerns to family members, relaxing by exercising with yoga or reading, were some of the methods proposed for dealing with sleep disorders at home [77]. 

## 6. Conclusions

Our study emphasized the need for supporting HCPs’ mental health and sleep, especially in those treating COVID patients and working in the primary care. Poor sleep is associated with a higher risk of developing mental illness such as generalized anxiety and depressive symptoms especially during a pandemic. Improving the sleep of HCPs is essential during the COVID-19 pandemic. The national health systems should implement effective strategies for the early identification of various risk factors and accurate recognition of sleep disorders and mental distress. The provision of effective management strategies in a timely manner is essential for the protection of HCPs and also of their patients [74,75,76]. Reasonable working schedules that allow the appropriate recovery of HCPs and relaxation techniques such as mindfulness and emotional support are essential [78]. On the other hand, there is still no longitudinal investigation assessing the duration and intensity of sleep disorders and psychological distress in medical staff. It is very important to evaluate the long-term effects of the pandemic and the effectiveness of the implemented supporting strategies. Additionally, it is necessary to assess the different outcome measures across different time points, the effect of vaccination or of other possible treatments for COVID 19, as the impact of the pandemic on mental health and sleep may change over time.

## Figures and Tables

**Table 1 healthcare-10-01395-t001:** Differences between socio-demographic characteristics, mental health indices and sleep quality of al the responders of the 1st and of the 2nd wave. (Mean ± s.d.).

	Wave 1 *n* = 574	Wave 2 *n* = 514	*p*
Age, years	45.5±11	45.1±8.2	0.4
Gender (males%)	47.4%	45.3%	0.39
Married	63.4%	64.2%	0.61
Children	33.6%	36.8%	0.4
Smoking	32.9%	33.1%	0.3
Physician	79.2%	81.8%	0.75
Pathological specialty	58.2%	59.0%	0.87
Surgical specialty	32.7%	30.9%	
Laboratory specialty	9.2%	10.1%	
Nurse	8.9%	10.7%	0.41
Physiotherapist	3.3%	0.0%	0.001
Work in ICU	9.9%	11.5%	0.22
Treating COVID 19 patients	19.3%	46.4%	<0.001
HSI	2.44 ± 1.7	2.41 ± 1.6	0.8
DAR5	7.7 ± 2.6	9.1 ± 3.6	<0.001
DAR-5 > 12	8.5%	22.4%	<0.001
LS	5.01 ± 1.6	5.9 ± 1.6	<0.001
LS > 6	35.4%	51.3%	<0.001
SCI	26.4 ± 5.7	23.4 ± 6.2	<0.001
SCI < 16	6.7%	19.7%	<0.001
PHQ4	2.8 ± 2.3	4.5 ± 1.3	<0.001
Normal (0–2)	48.4%	26.9%	<0.001
Mild (3–5)	39.9%	41.0%	
Moderate (6–8)	9.6%	20.1%	
Severe (9–12)	2.1%	12.0%	
PHQ4 Anxiety	1.34 ± 1.21	2.2 ± 1.9	<0.001
PHQ4: Anxiety > 3	13.1%	33.1%	<0.001
PHQ4 Depression	1.53 ± 1.30	2.41 ± 2.1	<0.001
PHQ4: Depression > 3	16.9%	38.2%	<0.001

s.d. = standard deviation, ICU = Intensive Care Unit, HSI = Heaviness of Smoking Index, PHQ4 = patient health questionnaire-4, DAR-5 = Dimensions of Anger Reactions-5, LS = UCLA Loneliness Scale, SCI = Sleep Condition Indicator; Independent-samples-*t*-test for parametric and the Mann–Whitney-U-test for non-parametric variables were used for continuous variables; Chi-square was used for categorical variables.

**Table 2 healthcare-10-01395-t002:** Differences between socio-demographic characteristics, mental health indices and sleep quality of the responders that were followed up during both first and second waves (Mean ± s.d.).

	Wave 1 *n* = 469	Wave 2 *n* = 469	*p*
Work in ICU	9.2%	12.8%	0.08
Treating COVID 19 patients	21.%	48.6%	<0.001
HSI	2.3 ± 1.8	2.4 ± 1.7	0.78
DAR5	7.3 ± 3.3	9.2 ± 3.8	<0.001
DAR-5 > 12	7.6%	22.0%	<0.001
LS	4.9 ± 1.6	5.9 ± 1.9	<0.001
LS>6	34.1%	54.3%	<0.001
SCI	25.4 ± 3.2	23.7 ± 6.6	<0.001
SCI<16	6.1%	18.6%	<0.001
PHQ4	2.7 ± 3.4	4.4 ± 2.9	<0.001
Normal (0–2)	49.1%	26.2%	<0.001
Mild (3–5)	40.1%	42.0%	
Moderate (6–8)	8.9%	20.3%	
Severe (9–12)	1.9%	11.5%	
PHQ4 Anxiety	1.29 ± 1.26	2.08 ± 1.58	<0.001
PHQ4: Anxiety > 3	12.7%	32.8%	<0.001
PHQ4 Depression	1.47 ± 1.2	2.38 ± 1.60	<0.001
PHQ4: Depression > 3	15.8%	37.7%	<0.001

s.d. = standard deviation, ICU = Intensive Care Unit, HSI = Heaviness of Smoking Index, PHQ4 = patient health questionnaire-4, DAR-5 = Dimensions of Anger Reactions-5, LS = UCLA Loneliness Scale, SCI = Sleep Condition Indicator; Paired *t*-test or McNemar’s test were used for parametric and non-parametric variables, respectively. Chi-square was used for categorical variables.

**Table 3 healthcare-10-01395-t003:** Comparison of sleep-related variables of participants followed up during the 1st and 2nd wave.

		1st Lockdown (*n* = 469)	2nd Lockdown (*n* = 469)	*p* (Value)
What is the quality of your sleep during the pandemic compared to before that?	Worse	15.7%	32.8%	<0.001
	The same	71.1%	62.7%	
	Better	13.2%	4.5%	
The last 2–3 weeks how much of a problem were any episodes of awakening during your sleep?	No problem at all	43.2%	35.8%	<0.001
	Small problem	39.9%	37.7%	
	Moderate problem	15.7%	22.6%	
	Great problem	1.2%	3.8%	
Do you use sleeping pills?	I do not use	93.0%	92.3%	0.67
	I was using them before the lockdown	4.9%	4.7%	
	I started using them during the lockdown	2.1%	3.0%	
SCI: How long does it take you to fall asleep?	> 60 min (0)	4.9%	6.0%	<0.001
	46–60 min (1)	0.0%	13.9%	
	31–45 min (2)	0.0%	2.3%	
	16–30 min (3)	40.4%	31.8%	
	0–15 min (4)	54.7%	46.1%)	
SCI: If you then wake up during the night … how long are you awake for in total? (add up all the awakenings)	> 60 min (0)	2.6%	5.8%	<0.001
	46–60 min (1)	7.7%	0.0%	
	31–45 min (2)	0.0%	10.0%	
	16–30 min (3)	12.9%	16.0%	
	0–15 min (4)	76.8%	68.2%	
SCI: How many nights a week do you have a problem with your sleep?	5–7 nights (0)	5.9%	10.7%	0.002
	4 nights (1)	6.4%	8.1%	
	3 nights (2)	8.9%	12.4%	
	2 nights (3)	13.9%	15.6%	
	0–1 nights (4)	64.8%	53.3%	
SCI: How would you rate your sleep quality?	Very poor (0)	0.3%	0.4%	<0.001
	Poor (1)	6.3%	0.0%	
	Average (2)	24.6%	17.3%	
	Good (3)	0.0%	28.8%	
	Very good (4)	68.8%	53.5%	
SCI: Thinking about the past month, to what extent has poor sleep affected your mood, energy, or relationships?	Very much (0)	1.2%	2.6%	<0.001
	Much (1)	4.5%	13.7%	
	Somewhat (2)	17.6%	25.1%	
	A little (3)	20.7%	19.8%	
	Not at all (4)	56.0%	38.8%	
SCI: Thinking about the past month, to what extent has poor sleep affected your concentration, productivity, or ability to stay awake?	Very much (0)	0.5%	2.3%	<0.001
	Much (1)	5.0%	9.4%	
	Somewhat (2)	16.3%	25.5%	
	A little (3)	18.0%	21.4%	
	Not at all (4)	60.2%	41.3%	
SCI: Thinking about the past month, to what extent has poor sleep troubled you in general?	Very much (0)	0.7%	1.8%	<0.001
	Much (1)	3.6%	9.1%	
	Somewhat (2)	12.2%	26.2%	
	A little (3)	18.7%	20.9%	
	Not at all (4)	64.8%	42.1%	
SCI: How long have you had a problem with your sleep?	>12 months (0)	10.3%	11.3%	<0.001
	7–12 months (1)	1.2%	5.6%	
	3–6 months (2)	2.8%	4.9%	
	1–2 months (3)	11.1%	11.5%	
	<1 month/I do not have a problem (4)	74.6%	66.7%	

s.d. = standard deviation, SCI = sleep condition indicator; Chi-square was used for categorical variables.

**Table 4 healthcare-10-01395-t004:** Comparison between primary care HCPs with all the others participants during the 1st and 2nd wave.

	Primary Care HCP 1st Wave *n* = 236	All the Others 1st Wave *n* = 233	*p*	Primary Care HCP 2nd Wave *n* = 236	All the Others 2nd Wave *n* = 233	*p*
Age	43.4 ± 11.1	47 ± 12.8	0.003	43.4 ± 11.1	47 ± 12.8	0.003
Gender (male%)	38.5%	49.3%	0.01	38.5%	49.3%	0.01
Married	59.7%	72.1%	0.004	59.7%	72.1%	0.004
HSI	2.2 ± 1.6	2.6 ± 1.7	0.07	2.3 ± 1.6	2.3 ± 1.7	0.8
SCITOTAL	25.4 ± 6.2 **	27.4 ± 5.2 *	0.001	23 ± 6.6 **	24.3 ± 6.4 *	0.1
SCI < 16	11.5% $	5.1% ***	0.02	20.6% $	14.3% ***	0.15
PHQ4TOTAL	3.15 ± 2.3 *	2.5 ± 2.2 *	0.001	4.65 ± 3 *	4.1 ± 2.8 *	0.07
PHQ4-anxiety ≥ 3	14.8% *	9.3% *	0.05	36% *	29.3% *	0.15
PHQ4-depression ≥ 3	17.8%*	14.9% *	0.4	40.6% *	34.2% *	0.19
DAR5TOTAL	8 ± 2.6 *	7.2 ± 2.5 *	0.001	9.3 ± 3.9 *	8.8 ± 3.3 *	0.18
DAR5≥12	9.7%*	6.7%*	0.2	23.5% *	17.3% *	0.12
LONELINESTOTAL	5.2 ± 1.7 *	4.8 ± 1.6 *	0.009	5.9 ± 1.9 *	5.7 ± 1.85 *	0.5
LS > 6	38.4% ***	29.9% *	0.04	53.4% ***	52.7% *	0.9

HIS = Heaviness of Smoking Index, PHQ4 = patient health questionnaire-4, DAR-5 = Dimensions of Anger Reactions-5, LS = UCLA Loneliness Scale, SCI = Sleep Condition Indicator; Independent-samples-T-test for parametric and the Mann–Whitney-U-test for non parametric continuous variables were used to compare between the different groups of HCPs; Paired *t*-test or McNemar’s test were used for parametric and non-parametric variables, respectively in order to compare the same group during the two waves; Chi-square was used for categorical variables * comparison between 1st and 2nd wave *p* < 0.001 ** *p* = 0.002 *** *p* = 0.003, $ *p* = 0.05.

**Table 5 healthcare-10-01395-t005:** Multiple regression analysis examining the association between sleep quality and characteristics of the population of HCPs and separately of primary care HCPs followed during the 1st and the 2nd waves.

All Population (*n* = 4 69)						
Variables	Standardized β	t	p	R	R2	Adjusted R2
SCI 1st				0.58	0.34	0.32
age	−0.057	−1.25	0.2			
gender	−0.04	−0.85	0.4			
MARRIED	0.12	2.12	0.03			
Primary care HCP	0.092	2.045	0.04			
PHQ4	−0.49	−8.42	<0.001			
DAR5	−0.096	−1.75	0.08			
LONELINESS	−0.13	0.257	0.8			
Treating COVID patients	−0.057	−1.31	0.2			
SCI 2nd				0.616	0.38	0.36
age	0.11	1.29	0.2			
gender	−0.019	−0.228	0.8			
Married	−0.04	−0.67	0.5			
Primary care HCP	0.013	0.27	0.19			
PHQ4	−0.33	−4.9	<0.001			
DAR5	−0.17	−2.8	0.005			
LONELINESS	−0.23	−4.05	<0.001			
Treating COVID patients	−0.06	−1.16	0.25			
Only primary care (*n* = 236)						
SCI 1st				0.62	0.38	0.36
age	−0.04	−0.04	0.5			
gender	−0.03	−0.59	0.6			
PHQ4	−0.55	−6.85	<0.001			
DAR5	−0.065	−0.37	0.7			
LONELINESS	0.07	0.97	0.6			
Treating COVID patients	−0.08	−1.23	0.2			
SCI 2nd				0.59	0.35	0.32
age	−0.09	−1.13	0.3			
gender	−0.07	0.88	0.42			
PHQ4	−0.24	−2.34	0.02			
DAR5	−0.23	−2.5	0.015			
LONELINESS	−0.3	−3.6	<0.001			
Treating COVID patients	0.04	0.49	0.6			

PHQ4 = patient health questionnaire-4, DAR-5 = Dimensions of Anger Reactions-5, LS = UCLA Loneliness Scale, SCI = Sleep Condition Indicator.

**Table 6 healthcare-10-01395-t006:** Correlation of sleep quality with mood changes, anger and loneliness during the two waves. Analysis of all participants and separately for primary care.

	PHQ4	*p*	DAR5	*p*	LS	*p*	HSI	*p*
All Participants (*n* = 469)								
SCI 1st	−0.61	<0.001	−0.395	<0.001	−0.28	<0.001	−0.2	0.15
SCI 2nd	−0.46	<0.001	−0.45	<0.001	−0.45	<0.001	0.19	0.2
Only Primary Care (*n* = 236)								
SCI 1st	−0.55	<0.001	−0.38	<0.001	−0.31	<0.001	−0.115	0.2
SCI 2nd	−0.56	<0.001	−0.49	<0.001	−0.47	<0.001	0.08	0.47

PHQ4 = patient health questionnaire-4, DAR-5 = Dimensions of Anger Reactions-5, LS = UCLA Loneliness Scale, SCI = Sleep Condition Indicator, HSI = Heaviness of Smoking Index.

## Data Availability

The data presented in this study are available on request from the corresponding author. The data are not publicly available due to institutional ethics policy.

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
