# Peer review of "Sleep Disorders and Mental Stress of Healthcare Workers during the Two First Waves of COVID-19 Pandemic: Separate Analysis for Primary Care"

_healthcare, 2022, doi:10.3390/healthcare10081395_

Round 1
Reviewer 1 Report
I would like to thank you for submitting and give me the opportunity to review the manuscript entitled: “Are Sleep disorders and mental stress of Healthcare workers worse in primary care? Data from the Two first Consecutive Waves of COVID-19 pandemic.” The research topic undertaken by the authors is interesting in terms of the mental health side-effects of a pandemic on the mental health of healthcare workers. Nevertheless, some questions and concerns need to be answered and corrected before the formal acceptance of the manuscript.
A clear hypothesis is missing.
In your study, why you not consider any previous history of anxiety and/or depression episodes in the participants? Don't you think it might affect the results? Justify.
In table 2, question 1: “What is the quality of your sleep during the pandemic compared to before that?”, could the second wave data reflect feelings of sleep quality in the 6 months before rather than before the pandemic? Please explain.
In table 3, the * compare between 1st and second wave p<0.001, when there is no * it means that there is no comparison? but there is a p? A clarification is needed.
The discussion is underdeveloped. For example, and if I am not mistaken, the increased anxiety and depression expressed by participants in the second wave is not explained/justified. Revise and develop.
It is essential to improve the formatting and English style.
On the other hand, and due to the clinic importance of the study, a detailed explanation of how these results are important for different stakeholders such as healthcare professionals, patients, national health systems and scientist needs to be commented on in the discussion/conclusion sections.
And finally, some grammatical errors, such as unify test formats (PHQ4 or PHQ-4), the font size is different in some paragraphs of the different sections of the article, there is double spacing between many words and missing space between others. Review and correct.
Author Response
We would like to thank the reviewer for the comments that helped us to improve our manuscript. We will answer to the comments point by point.
Reviewer 1
I would like to thank you for submitting and give me the opportunity to review the manuscript entitled: “Are Sleep disorders and mental stress of Healthcare workers worse in primary care? Data from the Two first Consecutive Waves of COVID-19 pandemic.” The research topic undertaken by the authors is interesting in terms of the mental health side-effects of a pandemic on the mental health of healthcare workers. Nevertheless, some questions and concerns need to be answered and corrected before the formal acceptance of the manuscript.
A clear hypothesis is missing.
Answer to reviewer
We would like to thank you for your comment. The aim of our study was to assess the impact of COVID-19 on Healthcare Professionals’ ( HCPs’) sleep and mental stress in a population of northern Greece during the first two waves of the pandemic and compare possible differences between primary care HCPs with non primary care HCPs. It has been proven in many studies that during the new pandemic, HCPs presented a significant prevalence of psychological problems, as anxiety and depression, and also a high prevalence of sleep disturbances. We are not aware of any other study in Greece, and more specifically northern Greece, assessing these aspects until now. We believe that this was rather clear throughout our study. However, we improved several points of the manuscript in order to make it more clear to the readers.
We hypothesized that, after experiencing the first COVID-19 wave, HCPs’ levels of sleep quality together with psychological distress, especially anxiety and depression, anger and loneliness would further deteriorate and especially in primary care HCPs that are more exposed to COVID 19. We aimed to compare the same groups of HCPs during the two consecutive waves and evaluated if demographic and social factors, exposure to COVID 19 and psychological distress affected sleep quality during the first two waves with a separate analysis for primary care HCPs.
In your study, why you not consider any previous history of anxiety and/or depression episodes in the participants? Don't you think it might affect the results? Justify.
Answer to reviewer
We would like to thank you for your comment. There is evidence that the pre-existence of mental health disorders has been found to be a predictor of the development of subsequent mental health conditions and of course sleep disorders in HCPs. We agree that it would be ideal to have a history of psychological health status in all healthcare providers in several time points of their professional life; however unfortunately this does not exist in the Greek system for the evaluation of HCPs. HCPs are assessed for possible physiological problems if they work in public facilities only once and this cannot be assessed. In the questionnaire there was a question about history of psychiatric disease as depression (data not shown) with only 0.9% of HCPs answering that they have received treatment. Due to the low prevalence, we did not include the existence of previous depression in our analysis. It is true that the existence of previous history of anxiety and/or depression before the pandemic could affect our results but we did not have this specific question in our questionnaire. We have included this in the limitations of the study.
On the other hand, the interesting point of our study was that symptoms of depression, anxiety and anger (even if they could exist in some HCPs before the pandemic) increased during the pandemic and were worse in primary care HCPs.
In table 2, question 1: “What is the quality of your sleep during the pandemic compared to before that?”, could the second wave data reflect feelings of sleep quality in the 6 months before rather than before the pandemic? Please explain.
Answer to reviewer
Thank you for your point. As we describe in the methods we have tested the questionnaire in a pilot study conducted from a sample of 30 HCPs working in G Papanikolaou Hospital before the initiation of data collection in both waves. We concluded that it was rather clear to all the participants that the question reflected the quality of sleep before the pandemic and not compared to the previous questionnaire 6 months ago.
In table 3, the * compare between 1st and second wave p<0.001, when there is no * it means that there is no comparison? but there is a p? A clarification is needed.
Answer to reviewer
In Table 3 (now table 4) we present the comparisons between different variables evaluating sleep or mood changes of primary care HCPs with all the others HCPs, with a separate analysis of the participants followed up during 1st and 2nd wave.
Apart from the differences between groups, when the sleep quality, depression, anxiety, anger and loneliness of each group of HCPs, i.e primary care HCPs or all the others, were compared between the two waves significant aggravation was found in almost all variables and in both groups. We have made comparisons for all the variables. The * compare between 1st and second wave means p<0.001, **means p=0.002 *** means p=0.003, $ means p=0.05. If there is no sign, the comparison is not significant (p>0.05)
The discussion is underdeveloped. For example, and if I am not mistaken, the increased anxiety and depression expressed by participants in the second wave is not explained/justified. Revise and develop.
Answer to reviewer
Thank you for your comment as we have focused the discussion of this paper especially on sleep disturbances. Please see the revised discussion of our manuscript where the increased anxiety and depression expressed by participants in the second wave is further explained.
It is essential to improve the formatting and English style.
Answer to reviewer
Thank you for your comment we will try to improve in the revised version of the manuscript
On the other hand, and due to the clinic importance of the study, a detailed explanation of how these results are important for different stakeholders such as healthcare professionals, patients, national health systems and scientist needs to be commented on in the discussion/conclusion sections.
Answer to reviewer
Thank you for your comment in order to improve our paper. As you will see in the revised manuscript, we have changed several parts of the discussion according to your suggestions
And finally, some grammatical errors, such as unify test formats (PHQ4 or PHQ-4), the font size is different in some paragraphs of the different sections of the article, there is double spacing between many words and missing space between others. Review and correct.
Answer to reviewer
Thank you for your comment. We have made several corrections in the revised version of the manuscript. Unfortunately, the formatting was not done by the authors but ‘automatically’ from the journal.
Reviewer 2 Report
Thank you for the opportunity to read this paper.
I agree that measuring mental stress and sleep disorder in healthcare workers is especially important during a pandemic situation. However, many studies have been published in this area, thus the authors should strongly justify the need for the current study. How would the results of this study differ?
It could be a plus point of this study as it compared mental stress and sleep disorder between the first and second waves of the pandemic. However, my concern is why the study measured differences between groups and not within a group, meaning why it included two groups of respondents, rather than measuring the variables over time in the same respondents. How do the authors confidently claim that mental distress and sleep disturbance are higher during the second wave when the variables were not measured during the first wave for the second group and vice versa for the first group? Could it be possible that mental stress and sleep disorders were even higher for the second group of respondents during the first wave?
More details are required in the methodology sections in terms of:
· Sampling approach and how samples were selected from each hospital – are they representative of the accessible population from each hospital?
· Instruments used in this study, such as the language of instruments administered and details of the instrument's validity and reliability. How the cut-off points of the scales were determined?
· In Page 3, lines 150-151: “The DAR-5 [31] is a 5-item self-report five-point Likert scale, ranging from 0-5” – to recheck this sentence, is it a five-point or six-point Likert scale?
Results:
· Provide details of the analysis performed on each variable using the legend below the tables, since the authors used a variety of tests: independent t-test, Mann-Whitney-U-test, paired t-test or McNemar’s test, chi-square and Fisher's exact test. Justification for the statistical test is necessary.
Discussion:
· It is noted that the authors have repeated the findings many times, which may cause boredom and confusion among the readers. I would suggest the authors be more systematic in presenting the results without repeating them.
· Supporting the findings with previous studies is essential but should be discussed more concisely and critically by contrasting the results with the current study and substantiating the possible reasons. Avoid repeating the same points or contents.
· The implications of the study should be based on the study findings and aims, especially since the current study compares two-time points. What do the findings highlight and how would they benefit healthcare professionals?
Limitations – need to be more specific, focusing on the study’s current methodology and findings.
Conclusion – include what new information does this study provide?
In general, check for typos, format and font size consistency.
Author Response
We would like to thank the reviewer for the comments that helped us to improve our manuscript. We will answer to the comments point by point.
Reviewer 2
- Thank you for the opportunity to read this paper. I agree that measuring mental stress and sleep disorder in healthcare workers is especially important during a pandemic situation. However, many studies have been published in this area, thus the authors should strongly justify the need for the current study. How would the results of this study differ?
Answer to reviewer
We would like to thank you for your comment.
To the best of our knowledge, this is the first study that has attempted to demonstrate the changes in sleep quality, mental distress, anger and loneliness in Greek Healthcare Professionals (HCPs) in relation to the evolving COVID-19 pandemic. To date, no longitudinal studies evaluating these factors on HCPs and especially primary care HCPs have been conducted in northern Greece. The aim of our study was to assess the impact of COVID-19 on HCPs’ sleep quality and mental stress in a population of northern Greece during the first two waves of the pandemic and compare in order to assess for possible differences between primary care HCPs with non-primary care HCPs. It has been proven in many studies that during the new pandemic, HCPs presented a significant prevalence of psychological problems, especially anxiety and depression, together with sleep disturbances. We are not aware of any other study in Greece, and more specifically northern Greece, assessing these aspects until now. We believe that this was rather clear throughout our study. However, we improved some points of the manuscript in order to make it more clear to the readers.
- It could be a plus point of this study as it compared mental stress and sleep disorder between the first and second waves of the pandemic. However, my concern is why the study measured differences between groups and not within a group, meaning why it included two groups of respondents, rather than measuring the variables over time in the same respondents.
- How do the authors confidently claim that mental distress and sleep disturbance are higher during the second wave when the variables were not measured during the first wave for the second group and vice versa for the first group? Could it be possible that mental stress and sleep disorders were even higher for the second group of respondents during the first wave?
Answer to reviewer
We would like to thank you for your comment as this point was not clear to the methods of our study. During the follow up of the HCPs in the second wave, the questionnaire asked the participants to declare whether they have completed the same questionnaire previously. From this question and by crosschecking with the demographics of the participants, we managed to have the same group of 469 HCPs that were followed up during both waves and measured the variables over time in the same respondents. For that, we are confident that the answers about the mental distress and sleep disturbance were worse during the second wave. In order to make it more clear, we have added this information to the methods and results of the revised manuscript. Also we have changed Table 1 presenting the data of all the HCPs that answered the questionnaire in the first and the second wave. In all the other tables we compare the same responders (n=469) followed up over time. We have added a table (Table2 in the revised version) comparing the answers of the 469 HCPs that were followed up during both waves and measured the variables over time in the same respondents
More details are required in the methodology sections in terms of:
- Sampling approach and how samples were selected from each hospital – are they representative of the accessible population from each hospital?
Answer to reviewer:
Thank you for your comment. The questionnaire of the study was sent to every employee of the hospitals and healthcare centers (physicians, nurses, physiotherapists, etc.) of both public and private sector of the province of Thessaloniki, Greece, via emails delivered from the local medical and nursing stuff associations during the first two epidemic waves of COVID-19. Unfortunately, we do not have detailed data on how this sampling could be representative from each hospital. However, for the comparison between waves there were no significant differences between HCPs’specialties. Additionally, we have added your point in the limitations of the study.
- Instruments used in this study, such as the language of instruments administered and details of the instrument's validity and reliability. How the cut-off points of the scales were determined?
Answer to reviewer:
Thank you for your comment. The questionnaires variability, validity and cut offs were implemented according to the published instructions of each one. Please see the methods of the manuscript and references 29-34.
- In Page 3, lines 150-151: “The DAR-5 [31] is a 5-item self-report five-point Likert scale, ranging from 0-5” – to recheck this sentence, is it a five-point or six-point Likert scale?
Answer to reviewer
Thank you for your comment. We have corrected to ranging from 1-5.
- Results:
Provide details of the analysis performed on each variable using the legend below the tables, since the authors used a variety of tests: independent t-test, Mann-Whitney-U-test, paired t-test or McNemar’s test, chi-square and Fisher's exact test. Justification for the statistical test is necessary.
Answer to reviewer
Thank you for your comment. We have provided the information needed below each Table.
- Discussion:
It is noted that the authors have repeated the findings many times, which may cause boredom and confusion among the readers. I would suggest the authors be more systematic in presenting the results without repeating them. Supporting the findings with previous studies is essential but should be discussed more concisely and critically by contrasting the results with the current study and substantiating the possible reasons. Avoid repeating the same points or contents.
The implications of the study should be based on the study findings and aims, especially since the current study compares two-time points. What do the findings highlight and how would they benefit healthcare professionals
Answer to reviewer
Thank you for your comment. We appreciate your suggestion and have changed several parts of the discussion of the revised paper.
- Limitations – need to be more specific, focusing on the study’s current methodology and findings.
Answer to reviewer
Thank you for your comment. We appreciate your suggestion and have added some more points to the limitations.
- Conclusion – include what new information does this study provide?
Answer to reviewer
Thank you for your comment. We appreciate your suggestion and have changed several parts in the revised paper.
- In general, check for typos, format and font size consistency.
Answer to reviewer
Thank you for your comment. We appreciate your suggestion and have changed several parts of the revised paper.
Reviewer 3 Report
This is an interesting study that examined the prevalence of sleep problems in health care workers in the two waves of the pandemic. However there are a few concerns:
The title may be misleading as quite a lot of analyses presented refer to the whole sample, and not just the comparison between the types of workers. The other point is regarding the independence of the two samples. If the two samples from the two waves consist of some overlap of individuals, why were independent samples test used? A better design would have been to follow-up the samples so that it could have been a longitudinal study where you could have observed true changes of the same participants in the two waves. This can be done with an anonymous online survey.
Author Response
We would like to thank the reviewer for the comments that helped us to improve our manuscript. We will answer to the comments point by point
Reviewer 3
This is an interesting study that examined the prevalence of sleep problems in health care workers in the two waves of the pandemic. However there are a few concerns:
The title may be misleading as quite a lot of analyses presented refer to the whole sample, and not just the comparison between the types of workers.
Answer to reviewer
Thank you for your comment. We appreciate your suggestion and have changed the title to: Sleep disorders and mental stress of Healthcare workers during the Two first Waves of COVID-19 pandemic: separate analysis for primary care
However, we would like to point out that an important part of the analysis of our study is dedicated to primary care HCPs. (Tables 4, 5, 6 in the revised manuscript and discussion)
The other point is regarding the independence of the two samples. If the two samples from the two waves consist of some overlap of individuals, why were independent samples test used? A better design would have been to follow-up the samples so that it could have been a longitudinal study where you could have observed true changes of the same participants in the two waves. This can be done with an anonymous online survey
Answer to reviewer
We would like to thank you for your comment as this point was not clear to the methods of our study. During the follow up of the HCPs in the second wave, the questionnaire asked the participants to declare whether they have completed the same questionnaire previously. From this question and by crosschecking with the demographics of the participants, we managed to have the same group of 469 HCPs that were followed up during both waves and measured the variables over time in the same respondents. For that, we are confident that the answers about the mental distress and sleep disturbance were worse during the second wave
In order to make it more clear, we have added this information to the methods and results of the revised manuscript. Also, we have changed Table 1 presenting the data of all the HCPs that have answered the questionnaire during the first and the second wave. In all the other tables, we compare the same responders (n=469) followed up over time. We have added a Table (Table 2 in the revised manuscript) comparing 469 HCPs that were followed up during both waves and measured the variables over time in the same respondents.
Round 2
Reviewer 2 Report
Dear authors, thank you for revising the articles and overall the general improvement is noted. However, some aspects must be considered:
First is the study design. It is mentioned in the revised version that data were collected from a large number of respondents twice in this study, thus please check the suitability of using the cross-sectional study design.
First is the study design. The revised version mentions that this study collected data twice from a large number of respondents. Therefore, please check the appropriateness of using the cross-sectional study design.
Second, please check the total number of respondents in this study as the data was collected twice from the same respondents (n= 469) and once from 150 respondents. So is the total number of respondents in this study 1088 (Page 6, line 177)?
Third, regarding the respondents' homogeneity test, please reconsider whether it should be a test between the first and second wave respondents or between the respondents who participated twice and once?
Others for readers’ clarity:
· If the same questionnaire is used at both points of data collection, is there a justification why the pilot study needs to be performed before collecting the actual data in both waves (Page 3, Line 103-104)? Or is there a typo in this sentence? Please check for clarity.
· Please restructure the sentence on Page 4, Line 156-157: “To separate parametric from non-parametric variables normality tests using the Kolmogorov-Smirnov test were performed”. Indicate correctly the purpose of normality tests.
· For binary and multiple logistic regression, please specify whether the dependent variables were categories in the analysis.
· Justify your decision to include all the respondents in the multiple logistic regression and not just those who participated twice in the study twice.
· Please provide the legends below the table more precisely and look at the examples in the published articles. It should avoid long sentences and should correctly indicate the statistical test used for the items. Use * only to indicate the significant results (p values).
· To check for typos in detail (eg. nursing stuff on Page 3, line 109).
All the best
Author Response
Dear reviewer, we would like to thank you again for your detailed comments in order to improve our manuscript.
Reviewer 2
Comments
Dear authors, thank you for revising the articles and overall the general improvement is noted. However, some aspects must be considered:
First is the study design. It is mentioned in the revised version that data were collected from a large number of respondents twice in this study, thus please check the suitability of using the cross-sectional study design.
Answer to reviewer:
Thank you for your comment.
‘Cross-sectional survey data are data for a single point in time. Repeated cross-sectional data are created where a survey is administered to a new sample of interviewees at successive time points. On the other hand, longitudinal studies involve information directly gathered in a survey following individual persons over time. In panel surveys, the same individuals are interviewed at multiple time points, referred to as waves. Reflecting both the cross-sectional (between individuals) and time-series elements, panel data are also referred to as ‘cross-sectional time-series’ data’ (https://dam.ukdataservice.ac.uk/media/455362/changeovertime.pdf).
We performed a cross sectional study; however, we have included the question about prior responders and have compared the same group of individuals over time. For that, our study could be characterized as cross-sectional time-series. We have made the appropriate alterations in the revised manuscript.
Second, please check the total number of respondents in this study as the data was collected twice from the same respondents (n= 469) and once from 150 respondents. So is the total number of respondents in this study 1088 (Page 6, line 177)?
Answer to reviewer:
Thank you for your comment. We agree that it could be confusing for the readers. So we have deleted the parts that all responders were studied altogether (as the 469 were the same responders).
Third, regarding the respondents' homogeneity test, please reconsider whether it should be a test between the first and second wave respondents or between the respondents who participated twice and once?
Answer to reviewer:
Thank you for your comment. We have chosen to test the healthcare professionals that responded in both first and second wave. We have presented the responses of all the responders of the first and of the second wave in Table 1; however the greatest proportion of our analysis was based in the comparison of the same 469 individuals over time. We have corrected the tables with the number of participants that were studied, in order to avoid this misunderstanding.
Others for readers’ clarity:
If the same questionnaire is used at both points of data collection, is there a justification why the pilot study needs to be performed before collecting the actual data in both waves (Page 3, Line 103-104)? Or is there a typo in this sentence? Please check for clarity.
Answer to reviewer:
Thank you for your comment. During the second wave we have added the question asking the participants to declare whether they have completed the questionnaire previously. For that we have performed a small pilot study during both waves.
- Please restructure the sentence on Page 4, Line 156-157: “To separate parametric from non-parametric variables normality tests using the Kolmogorov-Smirnov test were performed”. Indicate correctly the purpose of normality tests
Answer to reviewer:
Thank you for your comment. We have corrected the sentence in the revised manuscript.
- For binary and multiple logistic regression, please specify whether the dependent variables were categories in the analysis.
Answer to reviewer:
Thank you for your comment. For multiple regression analysis in the dependent variable we used SCI or PHQ4 as continuous variables. In the analysis of the SCI with cut off 16 we used binary logistic regression with SCI cut off 16 as categorical variable. We have made the appropriate corrections to the revised manuscript.
- Justify your decision to include all the respondents in the multiple logistic regression and not just those who participated twice in the study twice.
Answer to reviewer:
Thank you for your comment. In the multiple regression, we have included only the participants that participated twice in the study. We have added the number of participants in the table 5 in order not to confuse the reader
- Please provide the legends below the table more precisely and look at the examples in the published articles. It should avoid long sentences and should correctly indicate the statistical test used for the items. Use * only to indicate the significant results (p values).
Answer to reviewer:
Thank you for your comment. We have tried to limit the sentences in most of the Tables. Unfortunately, the reviewers asked to indicate the statistical test, for that it is indicated in the legends of the tables. In the instructions for authors we did not find this restriction for the use of only * for significant results.
- To check for typos in detail (eg. nursing stuff on Page 3, line 109).
Answer to reviewer:
Thank you for your comment. We have corrected the typos in the revised manuscript.
Reviewer 3 Report
The manuscript has improved - no further changes are recommended
Author Response
Dear reviewer, we would like to thank you again for your detailed comments in order to improve our manuscript.